# Explaining Adversarial Examples with Knowledge Representation

## Abstract

Adversarial examples are modified samples that preserve original image structures but deviate classifiers. Researchers have put efforts into developing methods for generating adversarial examples and finding out origins. Past research put much attention on decision boundary changes caused by these methods. This paper, in contrast, discusses the origin of adversarial examples from a more underlying knowledge representation point of view. Human beings can learn and classify prototypes as well as transformations of objects. While neural networks store learned knowledge in a more hybrid way of combining all prototypes and transformations as a whole distribution. Hybrid storage may lead to lower distances between different classes so that small modifications can mislead the classifier. A one-step distribution imitation method is designed to imitate distribution of the nearest different class neighbor. Experiments show that simply by imitating distributions from a training set without any knowledge of the classifier can still lead to obvious impacts on classification results from deep networks. It also implies that adversarial examples can be in more forms than small perturbations. Potential ways of alleviating adversarial examples are discussed from the representation point of view. The first path is to change the encoding of data sent to the training step. Training data that are more prototypical can help seize more robust and accurate structural knowledge. The second path requires constructing learning frameworks with improved representations.

## 1 Introduction

With the more widespread use of deep neural networks, the robustness and security of these networks have aroused the attention of both academic and industrial eyes. Among these adversarial examples is one of the most interesting as well as intriguing.

Since the discovery of adversarial examples in CNNs from 2013Szegedy et al. (2013), security and robustness has become a hot topic. Researchers have put efforts into finding out sources for adversarial examples and also developing methods for automatically generating these adversarial examplesGoodfellow et al. (2014).

Most these research focus on how certain perturbations lead to changes in decision boundaries. This paper discusses the origin of adversarial examples from a more underlying knowledge representation point of view. It provides a possible reason why adversarial examples exist for current networks and uses some experiments to prove this idea. Experiments also in some way show that adversarial examples can be derived from only the training data and totally network-independent. In addition, adversarial examples may be in more forms than the usual small perturbations. At last, possible ways to alleviate this issue are discussed.

### 1.1 Related Work

Current adversarial attacks have become a systematic procedure. Some algorithms have been developed to deliberately generate these kinds of adversarial examples Goodfellow et al. (2014); Moosavi-Dezfooli et al. (2016). After these examples have been generated, they can be injected back into the model to skew the classificationPapernot et al. (2017). This can even serve as a universal attack model for other machine learning techniques.

Some adversarial example generation techniques arise from the properties of neural networks themselves and are dependent on model architectures. Most of this kind of work has been done on image classification tasks like handwriting recognition (MINST dataset) and object recognition (ImageNet). However, from related research work these years, it has been widely recognized that there are even more machine architectures vulnerable to adversarial attacking other than neural networksPapernot et al. (2016).

More recently, some other research has shown that adversarial examples maybe more widespread in our physical worldKurakin et al. (2016). And there even exist universal perturbationsMoosavi-Dezfooli et al. (2017b;a) for a certain neural network that can generate examples that are both universal and adversarial. A recent paper further shows the existence of single-pixel attacks on image classification tasksSu et al. (2017).

Opposite to the attack techniques are the defense techniquesPapernot et al. (2015); Lu et al. (2017a). Some research has also been on this area. The most straightforward idea for defense is including some adversarial examples as inputs of the training set and let the neural network also learn what adversarial examplesTramèr et al. (2017a) are like.DeepDefenseYan et al. (2018) incorporates an adversarial perturbation-based regularization into classification objectives. A quite optimistic view comes from the research on multi-camera view indicating adversarial examples can be constrained by taking inputs of an object from different angles of viewLu et al. (2017b). And this multi-camera design aims to prove that adversarial examples may not be obvious for autonomous driving which must involve multiple cameras and scaling. However, some research shows that adversarial examples can directly work on image and scene segmentationsMetzen et al. (2017); Xie et al. (2017).

Other researchers also put many efforts into exploring the underlying reasons for these adversarial examples. The most direct one would be the linear vibration caused by vector computationGoodfellow et al. (2014); Tramèr et al. (2017b). This is the cause given by the fast gradient method paper. A more recent trend is that researchers try to use geometrical and topological methods to explore the perturbation effects on high-dimensional decision spaceWang et al. (2016); Tanay & Griffin (2016); Tramèr et al. (2017b); Liu et al. (2016). All these research shows a trend that people are more and more eager to undermine the principles of deep neural networks and extend their applications.

### 1.2 GENERAL IDEA

Even though we still have no clear idea how eyes and vision of human beings actually work on the neural level, human eyes are more error-resistant. It is not easy to see such kind of adversarial examples intuitively. Most of these kinds of examples are generated by carefully designed algorithms and procedures. This complexity to some extent shows that adversarial examples may only occupy a small percentage for total image space we can imagine with pixel representations.

To the best of our knowledge, this paper should be the first one that discusses adversarial examples from the internal knowledge representation point of view. The rest of this paper is organized as follows:

- The representation section gives a formal description of the explanation and illustrates why adversarial examples exist for current architectures.
- The experiment section proposes a one-step distribution imitation procedure to do some distribution imitations. Experiments show that simply by distribution imitation on a training set without any knowledge of the classifier may still lead to obvious impacts on classification results using the network. The last discussion section concludes the paper and discusses some potential solutions from knowledge representation perspective.

## 2 KNOWLEDGE REPRESENTATION OF NETWORK

When people think of an object, the object usually can be depicted in a quite abstract way. For example, when asked what is a car, we can think of a vehicle with several wheels and a driver steering in it. Even though for one task, there could be a large number of neurons involved, at least on high-level the encoding of the information should still be in a sparse way. For computers, the modern machine also set sparsity and abstraction as goals to seek. However, due to the lack of

accurate knowledge of optimal representations, current deep networks actually choose an alternative way with redundant parameters to depict tasks and objects.

For human beings, both abstraction and sparsity can be achieved with hierarchical storage of knowledge. This can be similar to object-oriented programming. We can think of an object with its attributes. Going back to the object recognition tasks, human beings can actually learn from one or a few prototypes for a certain object and know how this prototype can be transformed into different plausible forms. As a result, when recognizing an object, we are actually recognizing the object as well as how it is transformed from the prototype of this object to what we see.

Current multi-layer neural networks are partly enlightened by the hierarchical ideology. Even though throughout these years, the detailed architectures of deep neural networks have evolved a lot so as the accuracy, the most fundamental one is still the AlexNetKrizhevsky et al. (2012).

There are many layers but two main phases in this network for image classification tasks. The first phase mainly uses convolutional layers to extract local features from different scales. The second mainly uses fully connected layers to add up local features from the first phase with weights to construct a higher-level image entity. At last outputs from the second phase go through a Softmax classifier and give out probabilities for each possible class.

As described above, the architecture of AlexNet is quite intuitive. However, there still exists a great gap between abstract and sparse representation from human beings and neural networks. An ideal execution procedure can be simulated here to see how neural networks actually represent knowledge through the process.

Consider the two-step procedure of extraction and transformation. We can define the output of a network as: $output = T \cdot X$, where $X$ refers to the extracted part after the convolution and pooling part and $T$ refers to the transformations.

For a given network, different inputs can have different extracted patterns of $X$, here define the extracted $X = [X_1, X_2]$ consisting of two parts. Correspondingly, different inputs can activate different parts in the transformation, so we can have the output denoted as:

$$output = T \cdot X = [T_1, T_2] \cdot [X_1, X_2]^T \tag{1}$$

We can have inputs from the same class(same label). Some extreme cases can be considered here. Suppose for the first input, it is only extracted to be $[X_1, 0]$. For this input:

$$output1 = [T_1, T_2] \cdot [X_1, X_2]^T = T_1 \cdot X_1 + 0 = T_1 \cdot X_1 \tag{2}$$

In the same way, there could exist a second input that is only extracted to be $[0, X_2]$(for example, symmetric data). For this input:

$$output2 = [T_1, T_2] \cdot [X_1, X_2]^T = 0 + T_2 \cdot X_2 = T_2 \cdot X_2 \tag{3}$$

After the first input, parameters for the transformation $T = [T_1, 0]$ can be determined in the form of the extracted input.

For the ideal condition, the classifier should give the largest value for the correct class and very small but equal values for other classes before the softmax gives out the final probability distribution. The extreme case for the ideal softmax is to give positive infinite for the correct and negative infinite for others. So the outputs discussed above should be a fixed value when two inputs belong to the same class(they have the same output results).

Similarly, in the second step, given the same output result, there should be:

$$T_2 \cdot X_2 = T_1 \cdot X_1 \tag{4}$$

$$T_2 = (T_1 \cdot X_1) \cdot X_2^{-1} \tag{5}$$

Now the transformations are denoted by the first two inputs and we are given a third input and its output pattern can be denoted as $X_3 = [A \cdot X_1, B \cdot X_2]$, where $A, B$ are both matrices.

Put this back into the learned system of $output = [T_1, T_2] \cdot [X_1, X_2]^T$, we can get the output result:

$$output3 = [T_1, T_2] \cdot [A \cdot X_1, B \cdot X_2]^{-1} \tag{6}$$

$$= [T_1, (T_1 \cdot X_1) \cdot X_2^{-1}] \cdot [A \cdot X_1, B \cdot X_2]^{-1} \tag{7}$$

So the system actually processes the test input into:

$$output3 = T_1 \cdot A \cdot X_1 + (T_1 \cdot X_1) \cdot X_2^{-1} \cdot B \cdot X_2 \tag{8}$$

That actually indicates an interesting fact that what current neural networks have extracted at last are not only local features with combinations but also a weighted sampling of all training inputs. The visualization techniquesZeiler & Fergus (2014) can help show some elements of truth in this aspect.

We can use the DeepDream tool to visualize individual feature channels and their combinations to explore the space of patterns learned by the neural networkMordvintsev et al. (2015).

The tool returns images that strongly activate the channels within the network to highlight the features learned by a neural network. Nine random channels are chosen, they are 'balloon', 'container ship', 'grey whale', 'aircraft carrier', 'ashcan', 'radio', 'trolleybus', 'revolver' and 'passenger car'.

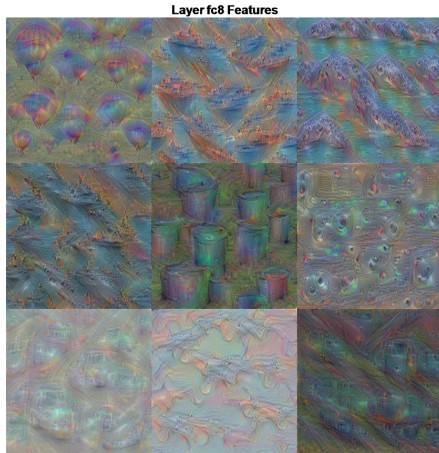

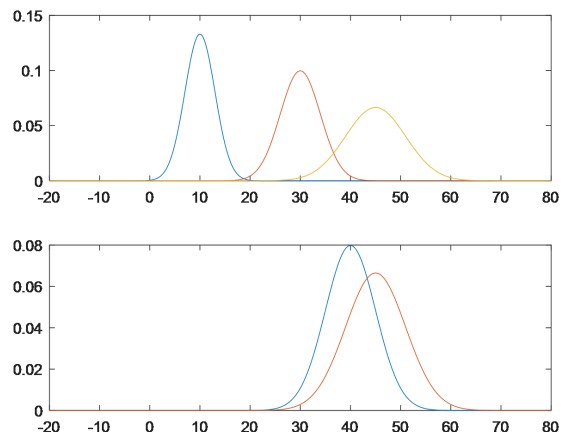

Figure 1: Visualize last layer with Deep-Dream.

Figure 2: Independent and hybrid representations.

It can be seen in Figure 1 that in the last layer with corresponding labels, the objects can still be recognized from their round and very similar repeating structures. For relative complex objects like 'container ship' or 'revolver' they can only ensemble similar shapes that are obviously overlapping with each other. This, in some way, proves the point that the knowledge storage and representation of current neural networks are not exactly sparse prototype based. Current networks actually use a hybrid distribution of all its prototypes from training and possible transformations to represent a certain class of object.

And the accumulation of these makes the space distances between different types of objects decision boundaries become smaller. When they become small enough, even small perturbations become possible to lead the classifier into a wrong result if imposed in certain directions. That may be the origin of adversarial examples.

The Figure 2 above provides a simple description of this idea. The first one represents the way of differentiating transformations in the same class. One class includes two normal distributions(after transformations) N(10,3) and N(30,4). There is one simple explanation of why one prototype of a class is reasonable to be denoted as an object. When a human is learning a class prototype from a physical object, there are actually an unlimited number of images from different angles and views streaming into human brains. And a large number of samples from a same class prototype will be converged to be a normal distribution. The other class includes one normal distribution N(45,6).

In this representation approach, we can actually easily differentiate between these prototypes with representations. Even their mean values are different enough. However, as we have discussed, the representation in current neural networks is more likely to be in the second figures form when multiple prototypes and transformations are involved. We can see in the second figure of Figure 2,

the representation of the first class combines two subclasses together. From the properties of normal distributions, the resulting hybrid representation becomes N(40, 5). This becomes more similar to the second class and makes it more difficult to classify. The absolute difference in means decreases from 45-30=15 to the smaller 45-40=5. The classifier has a lowest precision it can recognize. If right now a perturbation is added to make this difference even smaller to a certain degree, the classification cannot be guaranteed any more.

In summary, human beings can detect different objects as well as their transformations at the same time. CNNs do not separate these two. This makes final layers extracting not pure objects but in reality, a probability density distribution of the objects and its different states after transformations. Adversarial examples can arise from this underlying form of hybrid knowledge representation by imitating the distributions of a target class.

## 3 EXPERIMENT

As discussed above, the hybrid knowledge representation may lead to unclear bounds facing samples with high-dimensional distributions. One way to depict high-dimensional distributions is using weights from PCA. Similar values of weights, especially the first most important values, means higher similarity in distributions. And higher similarity in distributions means higher probability of lying in overwhelming areas between different class of objects. This part first gives an example of how the distribution of an adversarial example may vary after the perturbation. Then more systemic experiments based on one-step distribution imitations are conducted completely regardless of the network. And these experiments show how distribution imitation can impact the classification based on hybrid storage.

### 3.1 A SAMPLE OF WEIGHT MODIFICATION

Before diving into the direct distribution imitations, an example is shown here to show weight variations from a more commonly used adversarial attack method. The classifier training is based on handwriting digits and gets a classifier with 99.8% accuracy. To undermine more universal properties from the adversary, Fast Gradient MethodGoodfellow et al. (2014) is used for generating adversarial examples and reduce the success rate to 45%.

Here a set of examples is shown in Figure 3. The first one is a successful adversarial one, which makes the classifier recognizes a number 7 as a number 9. The second one is an unsuccessful one in which the classifier could still recognize a number 1 correctly as a number 1.

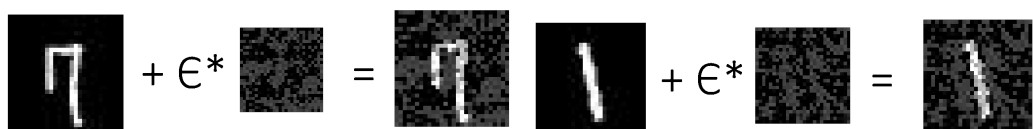

(a) 7 Adding Noise Recognized to be 9.          (b) 1 Adding Noise Still Recognized to be 1.

Figure 3: A set of examples of digit adversarial images.

Here we do PCA on all ten classes from 0 to 9. From these principal components feature space, we can infer what happens when the classifier classifies the data as 9 instead of 7. The adversarial example actually puts more weights on the relating principal components of 9 and this later misleads the classifier. This can be confirmed by the values of projections on normal and adversarial examples.

A simple metric that can be used here is the angle between projectionsFujimaki et al. (2005). It can be computed using the dot product and ArcCOS function. The angle on basis 7 between normal and adversarial is 0.2286 while on basis 9 is 0.1140. Smaller angle means higher similarity here. That is one view of classifying the modified image of 7 as 9.

Table 1: Projections of normal and adversarial images on Basis 7 and 9 (truncated)

| Projection | Score1 | Score2 | Score3 | Score4 | Score5 |
|---|---|---|---|---|---|
| 7 Nor | -193 | 108 | -39 | 582 | 274 |
| 7 Adv | -134 | 123 | 96 | 644 | 143 |
| 9 Nor | -199 | 364 | 26 | 812 | 3.7 |
| 9 Adv | -131 | 420 | 8.3 | 52 | 0.1 |

## 3.2 ONE-STEP DISTRIBUTION IMITATION

According to the distribution imitation idea discussed in section 2, we can actually take a one-step imitation to see whether the new image can deviate from the original classification result while still preserving the overall structure. We do this imitation by modifying the weight values from PCA.

Nowadays, PCA is mainly used for dimension reduction in the data pre-processing step. But it can be used as a rough classification method as well. When doing this classification work, first compute PCA on the training set and get coefficients and weights for training images. And then compute the weights of the test image using the coefficients. Choose the nearest neighbor according to a certain number of first weight values. And the class of this nearest neighbor is regarded as the output class of the test image.

Here, we are seeking a different goal. We know the label of the test image, but we want to modify this image so that a classifier trained from this training set will be more likely to misclassify the modified image.

Overall, we are using PCA weights to find the nearest different class neighbor and imitate the original image into similar patterns with this neighbor. A formal description of this procedure is given below.

---

**Algorithm 1:** Use PCA Weights to find the Nearest Different Class Neighbor and Imitate

**Data:** training image dataset $X = \{x_1, x_2, ..., x_n\}, TrainLabels$
**Data:** testing dataImage $y, yLabel$
**Result:** $x_{goal}, y_{modified}$

1   initialization;
2   $[coeff, score, mu] = pca(X)$;
3   $y_{score} = (y - mu) * inv(coeff)$;
4   $dist_{min} = +inf$
5   **for** $x \in X$ **do**
6     $dist = Distance(y_{score}, score[x])$;
7     **if** $dist < dist_{min} \cap yLabel \neq TrainLabels[x]$ **then**
8       $dist_{min} = dist$;
9       $x_{goal} = x$;
10     **else**
11       continue;
12     **end**
13   **end**
14   $y_{modified} = weightImitate(y, x_{goal})$;

---

The distance measure can be computed by the first certain number of most important weights which occupy a high percentage of total variance.

Given the procedure above, the next question is: how to imitate the goal image given the original test image? We implement this $weightImitate(y, x_{goal})$ function in two ways:

**Ratio Modification**    The weight value difference between the test image and the goal image it needs to imitate is calculated. And then add a ratio times this difference to the original weight value.

It can be denoted as $y_{score} = y_{score} + (x_{goal} - y_{score}) * ratio$. And then use this new weight vector to reconstruct the image.

**Weight Select Modification**   An alternative way is to choose some weight positions and assign the values from $x_{goal}$ to $y_{score}$ on positions chosen. That is $y_{score}[weight\_sel] = x_{goal}[weight\_sel]$. And then use this new weight vector to reconstruct the image.

Both ways discussed above are implemented in the experiment. Experiments are conducted on the first 5,000 images from cifar-10 dataset. There are two main reasons the experiment is done on 5,000 samples rather than more.

First, 5,000 (about 500 per category) should be enough to depict distribution differences for a limited number of dimensions required for comparison here;

Second, the very first step of the procedure discussed above involves the computation of PCA on the whole training dataset. In addition, the step of finding nearest different class neighbor needs to iterate through all samples. Too many samples will cause too much burden in computation.

One thing worth noticing is that there are three color channels in colored images, the distribution imitation execution is conducted by each channel. For purely gray-scale image dataset like MNIST, it would just do for one channel.

To test image classification results, a 24-layer network is used. It has 6 stacks of Conv, BatchNorm and Relu combinations together with two MaxPooling layers after the second and fourth stack. This is a network structure that is very common. This network in normal working condition has an accuracy of 85.1% on the test set of cifar-10.

Figure 4 shows the results from the fixed ratio(=0.2) method. All weight values from the original image are changed towards 0.2 times the distance between the goal and it. It reduces the accuracy from 0.851 to 0.756.

Some interesting results are shown here. Two dog images are classified as birds. A cat is classified as a truck. A horse is classified as a deer. Note that even though some results remain correct, their confidence probability reduce obviously.

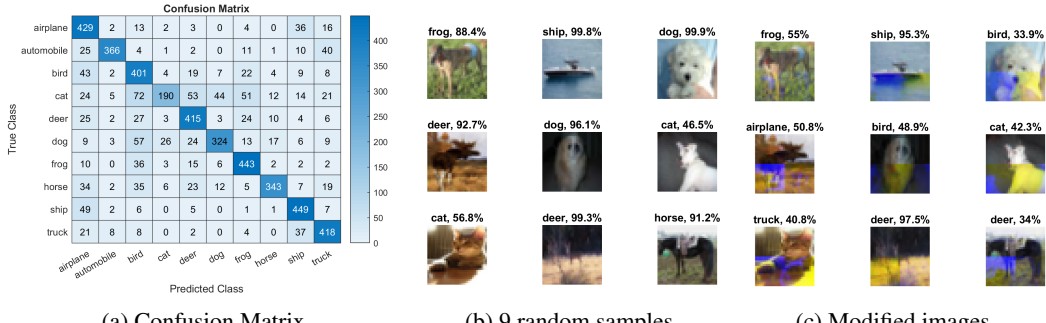

(a) Confusion Matrix.          (b) 9 random samples.          (c) Modified images.

Figure 4: Ratio=0.2 Weight Bias reduces accuracy by 10%.

Figure 5 shows the results from the row select modification method. Weight values on 50-100th positions are assigned with corresponding values from the goal. It reduces the accuracy from 0.851 to 0.532. Similarly, some obvious absurd classifications occur and even though some results remain correct, their confidence probability reduce obviously. Further tests are also conducted using an improved classifier adding ResNet connections. It has an accuracy of 0.588, which is not very different from the previous setting.

It can be observed that in both these two ways, the main structure of the original RGB image can be well preserved. But it differs from normal adversarial images in that it actually maximizes some color masking to the original image and this is sometimes visible. The position and visibility of this kind of masking depend on the magnitude and selection of weight changes. For the second method,

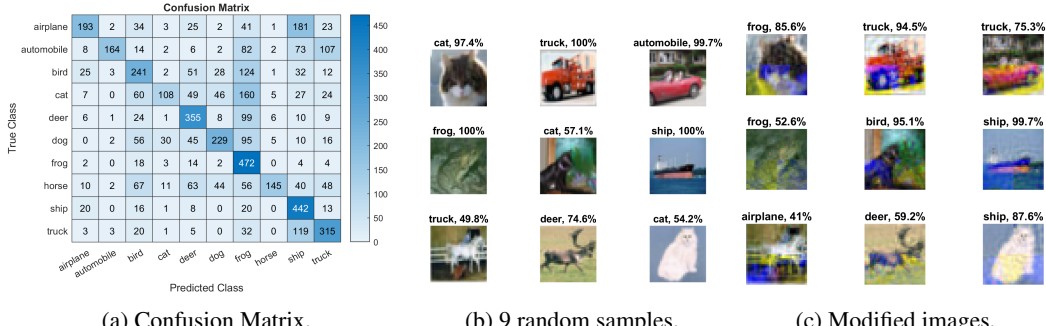

(a) Confusion Matrix.          (b) 9 random samples.          (c) Modified images.

Figure 5: Weight imitation on 50-100 components reduces accuracy by 32%.

if we chose more critical positions like weight 10-30, the main structure of the original image can be totally destroyed.

As RGB images have three color channels, the modifications from original images are sometimes not very visible. We do further experiments on the MNIST digit dataset with more flexible imitation settings. As we can see from experiments by two methods above, different weights have different impacts and the modification strength decides the overall result pattern.

**Ratio Piecewise Modification**   One hybrid way is to combine two modification methods above together. One 28*28 input image is divided into four 14*14 small patches. The dimension is 196. Instead of one fixed modification ratio for a number of scores, we apply modifications on scores from 10 to higher. For score 10-50, the ratio is linearly increased from 0 to a ratio level. And for scores higher than 50, the ratio is fixed at this level. This idea shown in Figure 6 serves as a piecewise function.

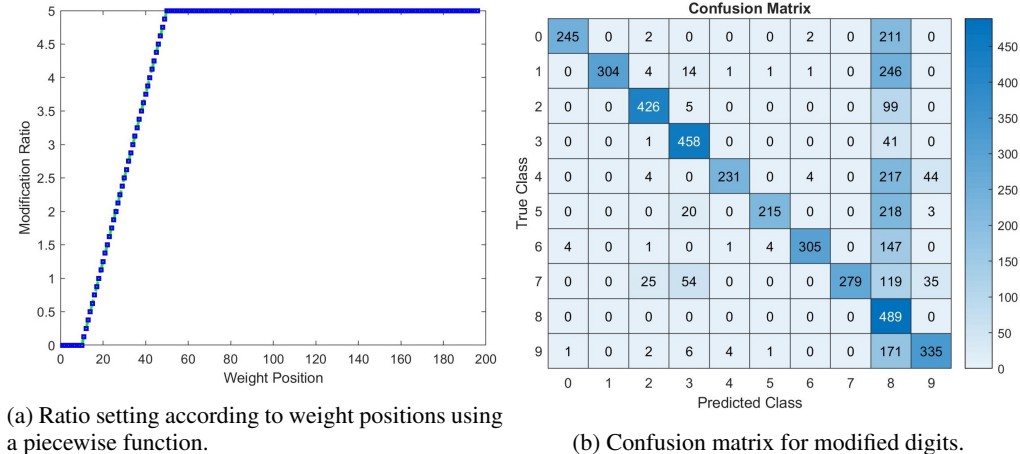

(a) Ratio setting according to weight positions using a piecewise function.

(b) Confusion matrix for modified digits.

Figure 6: Modification ratio and classification results.

Figure 7 shows the result when the selected highest ratio is 5. Under this condition the classification error increases from 1.04% to 34.20%. But this error rate requires relative strong modifications that are quite visible under black and white settings for MNIST dataset. One thing worth noticing is that the classifier is most easily fooled into recognizing the modified digit as 8. This concentration can be explained by the most widely connected and stretched for the digit '8' itself. This makes it the most distributed target in high dimension space and the ideal target for imitation in many cases.

We can see from Figure 7 that modified digit images are still recognizable for human eyes but they can successfully fool the neural network in our experiments. From a binary view, some noise is added to or removed from original images. The effect of this kind of perturbation is also supported

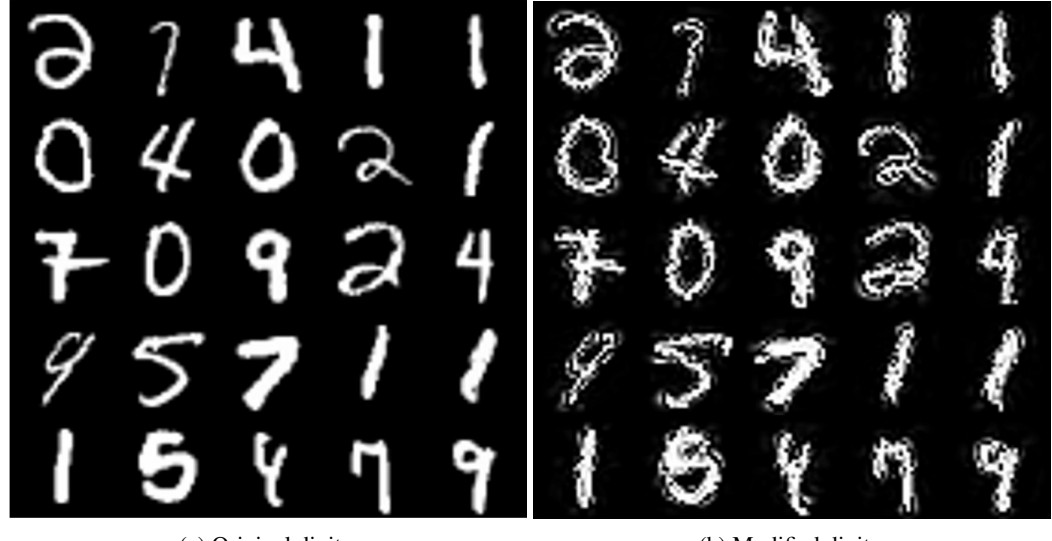

(a) Original digits.          (b) Modified digits.

Figure 7: 25 misclassified samples from the imitation that reduces accuracy by 33%.

by a recent paper on face recognitionWang et al. (2018). One main difference is that compared to RGB images with three color channels, modifications on black and white digits are more easily recognizable.

We conducted 50 sets of batch experiments. Each batch utilizes the first 500 samples from the test set. In these experiments, the modification ratio increases from 0.5 to 5. As shown in Figure 8, the classification accuracy gradually decreases from the initial 98.96% to 65.80% when the modification ratio equals 5.

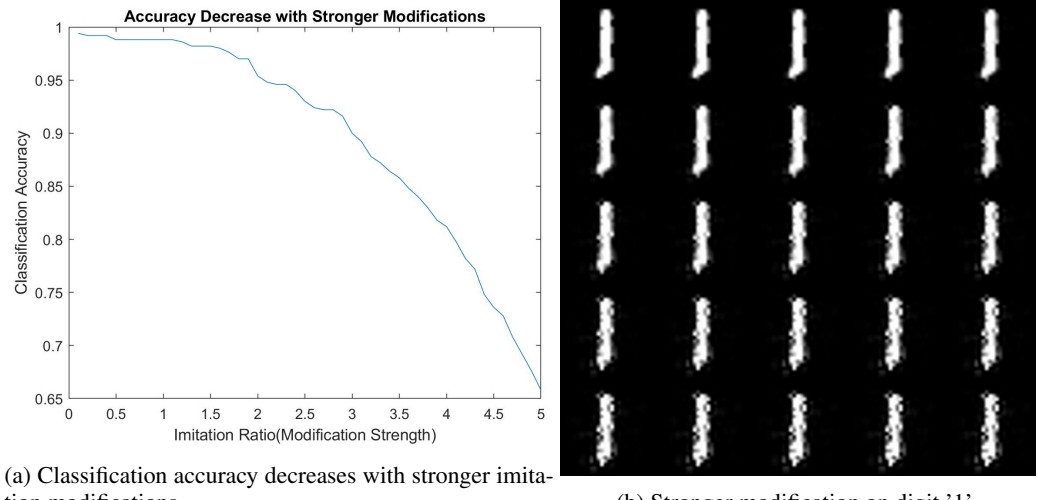

(a) Classification accuracy decreases with stronger imitation modifications.

(b) Stronger modification on digit '1'.

Figure 8: Experiment data for increased modification strength.From row 1 to 5 and col 1 to 5 modification ratio increases from 0.5 to 5.

It is worth pointing out that the procedure and experiments conducted here are mainly to show that adversarial images can be caused by distribution imitations on the dataset itself regardless of the network. It also shows that adversarial images can be in more forms than small perturbations.

This one-step distribution imitation procedure is not a formal method of adversarial attack. But considering the fact that it only needs to get access to the training data or even a part of the training

data, it can be conducted without any knowledge of the network. It still renders some danger if the attacker wants to misguide the classifier with many trials.

## 4 DISCUSSIONS

In summary, this paper discusses the origin of adversarial examples from an underlying knowledge representation point of view. Neural networks store learned knowledge in a more hybrid way that combining all prototypes and transformation distributions as a whole. This hybrid storage may lead to lower distances between different classes so that small modifications may mislead the classifier.

### 4.1 UNIVERSAL DISTRIBUTION IMITATION VS HIGH EFFICIENCY OF DEEP NETWORKS

The one-step distribution imitation procedure discussed imitates the nearest different class neighbor. Experiments conducted show that simply by distribution imitation on a training set without any knowledge of the network may still lead to obvious impacts on classification results. This also explains why adversarial examples can be universal.

Modified images are sometimes visible but still can keep original structures and information, this shows that adversarial images can be in more forms than small perturbations. Also, the robustness of classification is not only related to the classifier but also the dataset quality used for training.

One interesting question intuitively arising is while adversarial examples can be universal, why current deep neural networks show such high efficiency. A rational hypothesis is adversarial examples are rare from probabilistic view. A research diving into properties of layers Peck et al. (2017) shows lower bounds on the magnitudes of perturbations necessary to change the classification results. This also adds to the point that adversarial examples exist under strict conditions and cannot be generated in a pure random manner.

### 4.2 POTENTIAL SOLUTIONS FROM KR VIEW

**Dataset with More Concentrated Structural Info**  Without changing learning frameworks, this path focuses on changing what classifiers can learn from the very beginning.

For human beings, it is possible to learn a class of object from a single image sample. However, this is not the full story. Human beings are actually learning all the time throughout the process of using eyes to get visions. A hidden advantage for human vision is that humans have gained a good prior knowledge for various kinds of environments. This makes it easier for human beings to first judge out the relative distances from the object and more accurately seize a separate form of structural information.

In the same way, for a classifier learning from scratch, the training set with a monotonic background and a limited number of different object states can make learning space more concentrated and this in theory can help resist adversarial examples as early as from the training step. As far as we can see from current learning frameworks, this can be realized in two ways.

One way is to add pre-processing steps on datasets to make the data more compact instead of feeding data directly to the neural networks. A recent work shows that thermometer encodingBuckman et al. (2018) can help resist adversarial examples. This can be viewed as a frontend sampling step on the most significant information.

The other way is to make use of more prototypical datasets. Instead of giving one object in an image a single class label, it is also required to give a state description. This is equivalent to creating more subclasses and training the classifier to do further classifications. One potential dataset seeking this purpose is choosing learning prototypes from simplest toysWang et al. (2017). This proposed an egocentric, manual, multi-image (EMMI) dataset providing a more structured and denser sampling of viewpoints.

**Improved Representation of Learning**  The other path is to construct a better network representation. The success of deep networks points out the importance of hierarchical representations. However, recent research on neural network compression proves that knowledge representation in

current neural networks has a large number of shared and redundant parametersHan et al. (2015); Iandola et al. (2016). A deep network itself is still worth digging into.

According to the prototype and transformation representation model discussed, it is reasonable to separate these two kinds of knowledge in learning frameworks that involve detection and classification. For the current neural network for classification tasks, it is relatively simple to define a stable cost function and do the training globally. However, it we really want to separate these prototypes and transformations at the same time, it is almost impossible to define it as a single optimization problem. And state overlapping between different prototypes will make the final decision fuzzy.

In this way, we still need to face the dilemma that current hybrid representations in some way make adversarial examples inevitable. It might be necessary to design a new learning framework that represents knowledge in a more compact way. No matter what path to choose, we should still realize that current datasets or representation models are far away from human beings accumulation through years of vision usage. There is still a long way to go before machines can understand and represent vision information as good as or even better than human beings can do.

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
