# OpenReview forum: "Explaining Adversarial Examples with Knowledge Representation"
_ICLR.cc/2019/Conference_

### Official Review · AnonReviewer1 · 2018-10-31

**Rating:** 2
**Confidence:** 5

**Review:**

Strength:

The proposed approach is architecture-independent: the attack is constructed only from the dataset.

Weaknesses:

Paper is not sufficiently well written for a venue like ICLR.
Attack has very low success rate.
To the exception of Figures 4 and 5, many experiments are conducted on MNIST.

Feedback:

Experimental results show that the attack is able to degrade a classifier’s performance by inserting perturbations that are computed on the data only. However, there are no baselines included to compare adversarial evasion rates achieved here to prior efforts. This makes it difficult to justify the fairly low success rate. In your rebuttal, could you clarify why baselines were not used to offer comparison points in the experiments?

Furthermore, strong conclusions are made from the results despite the lack of supporting evidence. For instance, on P10, the attack is said to “also explains why adversarial examples can be universal.”. However, not only does the attack achieve less success than universal adversarial examples would (so it cannot explain all of them) but also does it not share any characteristics with the way universal adversarial examples are crafted. Drawing such a strong conclusion thus most likely needs a lot more supporting evidence.

Several directions would improve the content of the paper:

* Complete existing experimental results by being more systematic. For instance, in Section 3.1, measurements are only performed on one pair of MNIST images. Without studying a significant portion of the test set of two datasets, it is very difficult to draw any conclusions from the limited evidence.
* Perform a human study to have all perturbed images labeled again. Indeed, because of the ways images are perturbed here, it is unclear how much perturbation can be added without changing the label of the resulting input.
* Study how existing adversarial example techniques modify internal representations. This would help support conclusions made (e.g., about universal perturbations---see above).
* Rewrite the related work section to scope it better: for instance, Sabour et al. in Adversarial Manipulation of Deep Representations and Wicker et al. in Feature-Guided Black-Box Safety Testing of Deep Neural Networks explore adversaries operating in the feature space. This will also help build better baselines for the evaluation.

Additional details:

TLDR: typo in the first word
P1: The following definition of adversarial examples is a bit restrictive, because they are not necessarily limited to vision applications (e.g., they could be found for text or speech as well). “Adversarial examples are modified samples that preserve original image structures”
P1: The following statement is a bit vague (what is obvious impact referring to?): “Experiments show that simply by imitating distributions from a training set without any knowledge of the classifier can still lead to obvious impacts on classification results from deep networks.”
P1: References do not typeset properly (the parentheses are missing: perhaps, the \citep{} command was not used?)
P2: What is the motivation for including references to prior work in the realm of image segmentation and more generally-speaking multi-camera settings in the related work section?
P2: Typo in “ linear vibration”
P2: It remains difficult to make a conclusion about humans being robust to the perturbations introduced by adversarial examples. For instance, Elsayed et al. at NIPS 2018 found that time-constrained humans were also misled by adversarial examples crafted to evade ML classifiers: see Adversarial Examples that Fool both Computer Vision and Time-Limited Humans.
P2: Prior work suggests that the following conclusion is not entirely true: “Most of these kinds of examples are generated by carefully designed algorithms and procedures. This complexity to some extent shows that adversarial examples may only occupy a small percentage for total image space we can imagine with pixel representations.” For instance, Tramer et al. in ICLR 2018 found that adversarial subspaces were often large: “Ensemble Adversarial Training: Attacks and Defenses”.
P2: Others have looked at internal representations of adversarial examples so the following statement would be best softened: “To the best of our knowledge, this paper should be the first one that discusses adversarial examples from the internal knowledge representation point of view.”. See for instance, Deep k-Nearest Neighbors: Towards Confident, Interpretable and Robust Deep Learning by Papernot and McDaniel.
P3: Could you add pointers to support the description of human abstraction and sparsity? It reads a bit speculative as is, and adding some pointers would help relate the arguments made to relevant pointers for readers that are less familiar with this topic.
P3: What is the motivation for including the discussion of computations performed by a neural network layer-by-layer in Section 2?
P4: Given that saliency maps can be manipulated easily and are only applicable locally, it appears that Figure 1 is too limited to serve as sufficient evidence for the following conclusion: “This, in some way, proves the point that the knowledge storage and representation of current neural networks are not exactly sparse prototype based.”
P5: The error rate reported on MNIST is quite low (45%). Even using the Fast Gradient Method, one should be able to have the error rate be as high as 90% on a standard CNN.
P7: Would you be able to provide references to backup the following statement? “This is a network structure that is very common.”
P10: How does the discussion in Section 4.2 relate to the attack described in the submission?

---

### Official Review · AnonReviewer3 · 2018-11-02
**Very hard to read**

**Rating:** 3
**Confidence:** 2

**Review:**

**First of all, this paper uses 11 pages**
Submission instruction is "There will be a strict upper limit of 10 pages."

The readability of the manuscript should be improved.

I'm not convinced why Chapter 2 motivates Chapter 3. I think Ch. 2 and Ch. 3 are different stories.

---

### Official Review · AnonReviewer2 · 2018-11-05
**An interesting method, but the motivation should be clarified and the comparisons to the state-of-the-art should be improved**

**Rating:** 3
**Confidence:** 4

**Review:**

The paper discusses two ways of constructing adversarial examples (images) using PCA+knn in the input space. Compared to the litterature on adversarial examples, the modifications proposed by the authors are clearly visible to the human eye and the resulting images do not seem natural (see Figure 4 and 5). The authors acknowledge this difference between their work and the state-of-the-art (e.g., "Modified images are sometimes visible but still can keep original structures and information, this shows that adversarial images can be in more forms than small perturbations", Section 4.1), but it remains unclear why generating such images would be interesting in practice.

The algorithm for generating adversarial examples from nearest neighbors and PCA is reasonable. It seems simple and fairly easy to implement. However, it does not seem to be competitive with the current litterature for generating adversarial examples. An important point of the authors is that their method constructs "adversarial" samples without taking into account the specific structure of neural networks (more generally, without any knowledge of the classifier). This claim would have more practical impact if the method was shown to fool more algorithms/types of models than usual approaches (e.g., fast gradient sign). But there is no comparison to the state-of-the-art, so it is unclear in what situation the method should be interesting.

I found the motivation based on knowledge representation rather confusing, and I found no clear arguments for the PCA nor the k-nn approach. The write-up uses statements that are vague or not properly justified such as "For human beings, both abstraction and sparsity can be achieved with hierarchical storage of knowledge. This can be similar to object-oriented programming" (why is object-oriented programming relevant here?, is there any justification and formal statement for the first claim (e.g., a reference)?), "Neural networks store learned knowledge in a more hybrid way", "In summary, human beings can detect different objects as well as their transformations at the same time. CNNs do not separate these two." (there is no clear experiment proving that there is no "separation" because it is unclear what the DeepDream visualization of Figure 1 effectively proves). The equations do not really help (e.g., X_2 is supposed to be a vector if I understand correctly, what is X_2^{-1}?). Overall, I think the paper would gain a lot by making the motivation part more formal.

In conclusion, the authors seem to depart from the idea that adversarial examples should a) fool the network but b) still feel natural to humans. There is no clear motivation for generating such unnatural adversarial examples, and there is no clear application scenario where the algorithm would generate "better" adversarial examples than usual methods.

minor comments:

* please add spaces before \cite commands

---

### Meta-Review · Area_Chair1 · 2018-12-17
**reject**

**Confidence:** 5
**Recommendation:** Reject

**Metareview:**

The reviewers have agreed this paper is not ready for publication at ICLR.